# Regulatory Limits to Corporate Sustainability: How Climate Change Law and Energy Reforms in Mexico May Impair Sustainability Practices in Mexican Firms

**Antonio Lloret** [1,2,*], **Rogerio Domenge** [1,2] and **Mildred Castro-Hernández** [2]

[1]   Instituto Tecnológico Autónomo de México, México City 01080, Mexico; domenge@itam.mx
[2]   ITAM Center of Energy and Natural Resources, Mexico City 01080, Mexico; sgmil09@gmail.com
*   Correspondence: antonio.lloret@itam.mx

**Abstract:** This paper aims to show that sustainable behavior by firms may be impaired by regulatory restrictions. We challenge the assumption that regulation aimed at curbing greenhouse gas emissions (GHG) in the form of a target to meet the Country's GHG emissions commitments will promote sustainable corporations. We argue that, in fact, such regulation may impair sustainability practices because it creates unintended consequences. This paper tackles the efficiency of the institutional framework chosen through the lenses of the analytical themes of fit, scale, and interplay, then we use a systems dynamic approach to represent how regulation in the arenas of energy efficiency and GHG emissions reduction may withhold competitive business outcomes and corporate sustainability schemes. We exemplify and simulate a single regulation scheme: a clean energy target for firms; and found that as a result of such scheme, the system is dominated by negative feedback processes resulting in lesser outcomes that would be better tackled by firms not being subject to the restrictions imposed by the regulation.

**Keywords:** systems dynamics; corporate sustainability; Mexico energy reform; institutional analysis; implementation

## 1. Introduction

Energy sustainability, as defined by the World Energy Council, is the balance between energy security, social equity, and environmental impact mitigation [1]. This definition is aligned with the concept of sustainable development, which aims to find a balance between economic, environmental, and social systems. The development of energy schemes that are stable, affordable, and environmentally viable is not simple, and solutions to this problem are complex. The number of stakeholders involved is large and quite diverse; thus, any scheme to facilitate the balancing of these systems must consider that there are interconnections between the public and private productive sectors, governments, and regulators [1,2]. Different institutional frameworks must consider the demands of society, of companies, and of governments, including societal environmental concerns, in short: there are complex relationships within socioeconomic systems.

Since December 2013, Mexico has carried out an ambitious transformation process intended to end state monopolies in its energy sector. The constitutional amendments established new industry structures in oil, natural gas, and electricity. The aim was for competition to be introduced into the refined product and electricity markets, and for private investment to flow into various segments of these industries. The state would maintain ownership and control of subsoil hydrocarbon assets [3]. These reforms had been seen as positive for an emerging economy. Mexico's landmark energy sector

reform has the potential to transform and to grow the economy, and to attract billions of dollars of investment [4,5]. Nevertheless, economic growth comes at a price: a trade-off between economic growth and environmental protection, as carbon dioxide emissions soar and pollution becomes a severe problem amid economic progress and industrialization [6]. Businesses offering clean technologies will find growing opportunities as emergent economies work and invest to reduce environmental damage [7–9].

In December 2015, under the Paris Agreement, Mexico agreed on a Nationally Determined Contribution (NDC) that covers targets for both emissions of greenhouse gases (GHGs) and black carbon (BC). In its NDC, Mexico proposed to unconditionally reduce combined GHG and BC emissions by 25% below business-as-usual (BAU) levels in 2030 [10]. Mexico also proposed a 40% reduction of GHG and BC emissions by 2030, conditional on certain requirements for global agreement and international support. The GHG component of these targets translates to a reduction goal of 22% below BAU unconditionally, and 36% conditionally by 2030 [11]. One of the main institutions enacted was a new clean energy policy, its Energy Transition Law enacted in December 2015 [12], the law includes a clean energy target: 25% of electricity generation by 2018, 30% by 2021, and 35% by 2024, these targets have been relevant to shape GHG emissions trajectories [13,14].

While world leaders appreciate the Mexican contribution, these goals do not seem to have a clear grounding in economic productivity and competition. As of today, no sound analysis has been published that supports the NDC goals while also taking into account the economic growth of the country, nor on the technological innovation needed, nor on the funding that will be necessary, and it unfortunately has triggered some negative opinions from climate experts. Mexico would benefit from using strategies that have worked in developed countries, for example the EU Emissions Trading Scheme [15].

According to the Climate Action Tracker, a global independent consortium led by three research organizations tracking climate action, the April 2018 report rates Mexico's performance as: *"'Insufficient.' which means that Mexico's unconditional NDC commitment is not consistent with holding warming to below 2 °C, let alone limiting it to 1.5 °C as required under the Paris Agreement, and is instead consistent with warming between 2 °C and 3 °C."* The analysis suggests that Mexico will need to implement additional policies to reach its proposed NDC targets [16].

Whilst Mexico's progress in policy planning and institution infrastructure over recent years has been outstanding, the climate policy translates targets into strategies and plans, provides the institutional framework for implementation, but does not include concrete political instruments, making it impossible to quantify its direct effects [17,18]. The National Strategy on Climate Change (NSCC) is designed towards a long-term strategic development, and only provides very general guidance so implementation is of utmost importance [16].

The main problem lays not in setting ambitious targets but rather because the policy implementation and the institutional efficiency requires to both consider the direct responses associated with policy actions and the indirect responses that occur through complex relationships within socioeconomic systems [19], including the business sector [20]. The Mexican institutional transformation in the energy arena includes multiple policy instruments that will likely create indirect effects since system boundaries are uncertain and feedback among systems is complicated [21,22]. *"Complex systems may have both types of feedback loops, each with differing magnitudes and impact delays, creating nonlinearity and lag effects that can lead to unintended consequences that confound policymakers [23]."*

In order to address these complex relationships, we consider two important elements that are worth addressing for an effective implementation. First we aim at the national-wide institutional infrastructure and its effect on firms and second, the non-market alternative of corporate sustainability and how it must be considered to reach the NDC goals. We tackle these two elements to show that the Energy Transition Law aimed at curving GHG emissions in the form of a clean energy law, may prove to be ineffective and that its effects may impair the current corporate sustainability practices of Mexican firms [24]. In the next section we discuss through the lens of Young's analytical framework [25],

the interconnections between institutions and environmental change and the problems associated with a flaw implementation. Then, we address the non-market strategy of corporate sustainability to show that current regulation may impair firm's sustainable practices and finally we use a system dynamics approach to model a single institutional scheme to show and simulate what the feedback mechanisms of such policy scheme are and how improper implementation may only result in worse results with the regulation that without it. We close the paper with a discussion of results, conclusions, and an agenda for future research.

## 2. New Policies: The Complexity of Institutional Schemes

Institutional efficiency requires one to consider both the direct responses associated with policy actions and the indirect responses that occur through complex relationships within socioeconomic systems [9,26–28]. Young's analytical framework aims to understand the interconnections between institutions and global environmental change [27]. We argue that the current energy reform has many virtues but that improper implementation along with the complexity of the underlying socioeconomic system will results in not meeting the Mexico's NDC commitment. Young addresses the scientific questions of causality, performance, and design as the main questions to understand the institutional complexity to discuss the expected outcomes of such schemes [27,28]. Young argues that "*It is impossible to succeed in designing effective institutions without some understanding of the roles that these arrangements play as driving forces in the realm of human affairs* [28]"; thus, to understand it uses an analytical framework based on the domains of fit, scale, and interplay to suggest what ought to be achieved with the current institutional arrangement. The problem of fit centers on the proposition that there is a close fit between ecosystems and the institutional scheme chosen [27]. Scale deals with the levels at which phenomena occur in the dimensions of space and time, and determines whether the institutions designed will be sufficiently able to be scaled up or down to tackle inefficiencies. Interplay implies that other arrangements, both horizontally and vertically, will result in cross-scale interactions as a consequence of the politics of institutional design and management [25].

The scientific question of causality is concerned on how institutions influence human affairs. The reform and bylaws related to energy efficiency and climate change mitigation and adaptation stem from a consensus on what Mexico as a country ought to do as an emerging economy [11]. In Mexico, as in other developing countries, changes in extreme weather events are being observed. The Intergovernmental Panel on Climate Change, in its latest report [29], has confirmed that the climate change observed over the last 50 years is attributable to human activities, primarily the burning of fossil fuels and deforestation [30].

Mexico acknowledges that the attention of this phenomenon lies with all countries and states its commitment. It is clear that the consumption of goods and services is on the rise, and that this brings not only greater production and consumption of energy, but also a consequent increase in GHG emissions, which in turn rests on a greater use of natural resources and environmental pollution.

The scientific question of design aims to maximize institutional performance. The Energy Constitutional Reform, published on 20 December 2013 [31,32], has 23 articles that involve creating 11 laws and changing 12 existing laws. There are numerous amendments and new regulations set in place that will require adaptation and mitigation actions which firms directly or indirectly will have to consider in their sustainability strategies [33–36]. On 24 December 2015, the Energy Transition Law came into effect. The new law establishes a minimum requirement of clean energy use through the so-called "Clean Energy Certificates", or renewable energy certificates [12]. These "Clean Energy Certificates" are instruments issued by the Mexican Energy Regulatory Commission that evidence the generation of a specific amount of electric energy from clean energy sources. They are used to demonstrate compliance with clean energy consumption requirements as a portion of the total amount of energy consumed by the firm.

The scientific question of performance aims to assess the effects of the institutional scheme [25,27]. Mexico's Climate Change Strategy is conditional on the National Energy Strategy (NES). For firms, this

implies the adaptation and mitigation of their GHG emissions, a major challenge for companies. They must identify and generate information on emissions, reduction targets, innovation and development, and strategies. To achieve this, the General Law on Climate Change [37] requires companies that generate more than 25,000 tons of carbon dioxide equivalent ($CO_2e$) to report emissions in the National Register of Emissions (RENE). This standard includes a series of fines and penalties for companies that do not comply with their obligations. However, it is not clear how firms would measure certain strategies, such as spinoff or operations, logistics and distribution schemes, and subsidiaries, among others [19,38].

This instrument will inform regulators about energy consumption and environmental impact that may shape public policy. The problem is that RENE was voluntary for the first two years, 2016 and 2017, since the Energy Transition Law was enacted in late 2015 and compulsory thereafter. The first results of the compulsory scheme will be available as of the second semester of 2019 and it is expected that the first compulsory results go through a large review process that will require policy amendments. The targets and the emission results will be unsynchronized. Mexico's NDC commitment on clean energy included a clean energy target: 25% of electricity generation by 2018. Without clarity of RENE's results one can expect more uncertainty and firms may reduce their clean energy investments until there is a clearer implementation process based on sound information.

Causality, design and performance are the questions that drive the institutional complexities. To this end, we use the three analytical framework to understand the institutional complexity and why it may create unintended consequences.

## 2.1. The Analytical Framework of Fit

Young argues that an institutional arrangement that performs perfectly well in dealing with one environmental problem may prove to be a dismal failure when used in an effort to solve other problems. The 23 complex enacted laws, in some cases, are modified to attend new problems; however, the new regulations are not always clear about specific measures and do not create a roadmap to tackle the problems at hand.

Mexico's ambitious commitment has to come from several actors, including the government itself. With regards to firms, greenhouse gas reduction is targeted with (RENE): the tool that will provide timely, reliable, and verifiable information for assessing sector GHG emissions nationwide, and indicate whether companies are becoming more efficient in energy terms. Data considers that emissions by all private sectors in the Mexican economy account for anywhere between 20–30% of total GHG emissions [39]. A 30% reduction of the current trend line in firms will represent 10% of the total GHG emissions only in the best of cases, should the government not reduce other emission sources with public policy such as public energy production, transportation, agriculture, landfill, or land changes. This number seems to be disproportionate relative to a firm's emissions. Moreover, to aim for this goal in such a short time, then resources already committed to certain sustainability practices will potentially have to be transferred to different activities simply to meet these goals, with the unintended consequences of fewer sustainability activities being undertaken by firms [40–42].

## 2.2. The Analytical Framework of Scale

Scale concerns the levels at which phenomena occur in the dimensions of space and time across multiple levels of organizations. Problem complexities appear differently at different levels of scale produce special challenges for obtaining institutional fit [25,43]. Climate Change is clearly a scale-wise and temporary-wise problem. The institutional perspective includes an institutional design at the global, national, and local level across many geographies. Moreover, the schemes used worldwide needs not only to be adapted to national regimes but also to socioeconomic complexities within a nation state. As discussed in the analytical framework of interplay, there are several layers of institutions that cover a large geography making scale issues complex to address. Energy efficiency, as a potential solution, may come from the technology and infrastructure perspectives that include government

compromise, regulatory enforcement and market-based mechanisms [7,9,44]. Even with the current institutional arrangements in place Climate change is such a complex phenomenon, it is not clear whether the regulations will be capable of curbing emissions [21,40,45].

*2.3. The Analytical Framework of Interplay*

Institutions are interconnected at different levels, both vertically at cross scale interactions and horizontally at the same level. Although the energy sector is under federal jurisdiction, states and municipalities have responsibility for representing local interests, promoting opportunities for their people, and avoiding negative impacts on both society (considering the productive sectors) and ecosystems. Table 1, shows how different institutions are arranged at the three levels of government.

**Table 1.** Institutional schemes at the federal, state, and municipality levels.

| | JUDICIAL FRAMEWOR | PLANNING | INSTITUTIONAL ACTORS AND ARRANGEMENTS | FINANCING | INSTRUMENTS | EVALUATION | ENFORCEMENT |
|---|---|---|---|---|---|---|---|
| NATIONAL | General Law on Climate Change | National Strategy of Climate Change | • National System on Climate Change<br>• National Institute for Ecology and Climate Change<br>• National Congress | Climate Change Fund | • **Emissions Inventory**<br>• **Risk Atlas**<br>• **National Emission Registry**<br>• **Economic Instruments (fiscal, financial, market-based)**<br>• **Information System** | Evaluations Coordination NIECC | Sanctions |
| FEDERAL | | Special Programs | • Inter-secretarial Commission on Climate Change<br>• Climate Change Council C3 | Fund for Climate Change | Official Federal Standards (OMS) | Evaluations Coordination NIECC | |
| STATE | Existing state laws on the subject of Climate Change | State Programs | State inter-secretarial Commissions on Climate Change ICCC | Climate Change Funds and State Funds | • State Inventory of Emissions<br>• State Risk Atlas | State Program Evaluation Procedures | |
| MUNICIPAL | | Municipal Programs on the Subject of Climate Change | Association of Municipalities | Climate Change Fund and management of other resources | Vulnerable Municipalities Risk Atlas | Municipal Program Evaluation Procedures | |

Therefore, planning and coordination between the three levels of government and the private sector is required to prevent the most likely problems in the implementation of the reforms. The strategy should integrate international best practices, including sustainability objectives as well as competitive ones. For example, states should cooperate in implementing and monitoring standards through processes under their jurisdiction, without creating obstacles to business activities, while also ensuring compliance with social and environmental standards. Planning should seek to develop policies to exploit opportunities that promote synergies between firms and their local activities [25,27]. It is expected that the upcoming implementation of the energy reform, with the consequent changes for the environment, will lead to gains in the productive sectors of Mexico [46,47]. Most have anticipated costs, benefits, and investments, but at the same time, they have to consider making adjustments to their business models in order to be more efficient and sustainable [30,48–50].

From our previous analysis we can conclude that the complexity of the institutional infrastructure creates a number of challenges for firms. In the next section we address how corporate sustainability may be both an opportunity to meet the institutional performance as well as pose a challenge to the firm's sustainable strategies.

## 3. New Challenges for Firms: Corporate Sustainability within the Energy Reform

Nonmarket strategy refers to a firm's actions to improve its performance by managing the institutional context of economic competition [51,52]. The interest in this topic has existed for more than four decades [53] and has consolidated in recent years [36,54]. When faced with weak institutions or ineffective schemes, firms can create and appropriate value by adapting, augmenting, or transforming the existing institutional infrastructure. In such environments, managers must decide whether to adapt their strategies to the existing institutional environment, devote resources to improve it, or try to transform it altogether [55]. Corporate sustainability as a non-market strategy may be an approach that allows firms to meet ambitious goals.

Corporate sustainability derives from the broader sustainable development concept, and it is a construction parallel to Corporate Social Responsibility [3]. We characterize corporate sustainability as the ability of firms to create reasonable economic success with well-executed strategies that consider economic, environmental, and social constraints, maintaining social equity, environmental integrity, and economic success.

Corporate sustainability within energy reform implies impacts both internally and externally in economic, social, and environmental terms [53,56]. Knowing how and why these impacts are generated will allow firms to carry out strategies to undertake mitigation and adaptation strategies that will result in continuous environmental performance, focusing on the growing use of alternative fuels, reducing greenhouse gas emissions, and improving energy efficiency at the same time as maintaining their competitiveness [34,44,57].

For the case of many emerging economies, the idea is novel. In fact, in developed countries, the issue is at a mature stage regarding how and what to mitigate and adapt. In the case of Mexico, however, there are not as yet robust studies that can shed light on how, exactly, given the institutional infrastructure in place, companies will develop strategies and how the regulator should design public policies.

Some empirical studies have been conducted on environmental issues or environmental enforcement by adopting environmental management systems. Dasgupta et al. [58] who studied the effects of regulation and management policies at the plant level in the adoption of environmental practices, concludes that factors such as market management systems and subsidizing by the regulator when there is weak regulation can be useful tools for implementing complementary best environmental practices. Ruiz-Arredondo et al. [59] analyzed manufacturing enterprises in relation to the incentives to adopt subsidies, such as the Clean Industry Program ("National Environmental Audit Program"), the flagship program of voluntary regulation in Mexico. Their study concluded that regulatory pressure (implementation of the standard) and receiving fines or prosecutions initiated, as a result, the adoption of practices of stronger environmental management.

A broader study by Blackman et al. [60] used a sample of companies in the Business Information System (SIEM) of the Ministry of Economy which received fines linked to PROFEPA (the Federal Attorney for Environmental Protection). Their study supports the results that the main driver of participation is the threat of regulatory sanctions. Their study found that plants that sell their products in foreign markets, which are government suppliers, are relatively large and in certain sectors and states are more likely to engage in environmental management programs. Montiel & Husted [61] found, in the same sample as Blackman et al. [60], that the early adoption of voluntary programs in Mexico is further explained by access to international markets and the possibility of obtaining relevant information regarding the industry associations to which they belong.

On the flip side, analyzing the Porter hypothesis; Ambec [7] found that firms have higher incentives for establishing environmental innovation and foster competitiveness when the regulations are flexibles. Policies designed to go passed compliance through incentives can help reduce costs at the same time *"policy should strive to be win-win compatible. This speaks in favor of policies that provide incentives to innovation, are stable and predictable, make use of suitable transition periods, and focus on end results rather than means, and economic policy instruments [7]."* Market pressure for reducing GHG emissions is a

determinant for implementing a reduction strategy showing that corporate sustainability comes from within the industry and not from following regulations [38,62].

Several strategies focusing on differentiation of products, access to financing or cost reduction tend to be more effective because they focus on corporate sustainability rather than focusing only on following regulations. Having incentives to reward those leaders who follow sustainability acts do not force everyone to follow [63]. Another study indicates that firms focusing on emissions trading strategies does not directly reduce $CO_2$ emissions they do so trying to minimize their economic exposure. When companies invest in one action rather than on multiple strategies, they normally do so expecting higher returns. In comparison, when they are expecting lower emissions they focus on a combination of strategies despite these having different expected returns [64].

When following international negotiations; firms anticipate regulations that differ per region, there are uncertainties when implementing subsidies; having preferential loans for investment of clean technologies have created expectations for a wait-and-see period, that could jeopardize previous investments on proactive strategies [34]. Cadez's study [44] showed how an increase in eco-efficiency can be the trigger the reduction of costs by lowering pollution. Having different methods and frameworks to measure the quality of management improves the competition by having incentives though disclosure. Helping show there is a commonality in the management and in commitment, driving companies to a continuous improvement [19].

In a study made to identify the different motivations for having an ecological response; Bansal [33] was shown that competitiveness was one of the ultimate motivations by aiming to reduce costs and risks that these would have from noncompliance to the regulations. However; the motivations came more in the form of meeting the standards than actually exceeding them. The respondents of the study were cautions of the external constrained to avoid fines, penalties, sanctions, or bad publicity. Finally, Aigner & Lloret [65] generated information on various aspects related to the adoption of practices of environmental sustainability and the responsibility for decision-making regarding environmental challenges. The study found that Mexican firms have numerous drivers to invest in environmental activities, and that these firms are investing at increasing rates, suggesting that firms are in an early adoption stage.

As the energy reform and climate change laws were designed and enacted, the position of most of the Chambers of Commerce in Mexico has been positive but cautious [12,37]. Chambers such as the National Air Transportation Board and the Mexican Chamber of the Construction Industry consider that the reform can benefit their sectors in reducing the prices of their products, making them more internationally competitive. Companies will be able to sell their surplus generated electricity to third parties [66]. Many industries expect that the energy reform will bring new investments to their companies and some are developing projects that will benefit from the energy reform, while some others are wary that the implementation stage and enforcement is still lacking [46,67].

Among the chambers awaiting investments are the Mexican Chamber of Construction Industry (CMIC) Confederation of Industrial Chambers of the United Mexican States (CONCAMIN), National Association of Plastics Industries (ANIPAC) National Chamber of Sugar and Alcohol Industries (CNIAA), National Chamber of the Industry of the Iron and Steel (CANACERO), National Chamber of Fishing and Aquaculture Industries (CANAINPESCA) and the National Chamber of Industry Development and Promotion of Housing (CANADEVI). The expectation of all these chambers is that energy efficiency will result in less use of natural resources, lower energy consumption, and a decrease in GHG emissions.

Energy reform will trigger the dynamic that will allow Mexico to have the variety and quantity of energy that the country will require to meet its needs, achieve surplus exports, and compete in international markets [46]. The World Business Council for Sustainable Development (WBCSD) provides tools and protocols such as GHG Protocol or Sustainable Forest Finance for measuring the impacts of operations for implementing sustainability into corporate strategies. The WBCSD approach consists on believing *"that business must take the lead in identifying business solutions to tackle climate*

*change and the enabling conditions required. We envisage a portfolio of many solutions where companies—based on relevance, skills and leadership—can engage and help create action at scale. WBCSD does not believe that CEOs should be asked to commit their companies to emission reduction targets without substance. Instead, companies should contribute by implementing enabled sustainable solutions in their own operations, products and services as well as in their supply chains* [68]." The best practices expected for a company to engage with a low-carbon economy include having salient information to understand the facts with which to work from, focusing on the opportunities for a new climate economy when addressing the climate change challenges [69]. This would eventually follow the intergovernmental process of having the UN FCCC lead intergovernmental process and at the same time a business engagement through the private sector and civil society [10,68].

As we have argued, regulation in itself will not compel companies to meet the GHG emissions reduction targets and may impair current practices that are of utmost importance. In the next section, we address a single regulatory scheme, the Energy Transition Law, in a systems dynamic framework to show that regulatory constraints have unintended consequences in the non-market strategy of corporate sustainability in firms as a result of the feedback mechanisms that occur in complex socioeconomic systems.

## 4. Unintended Consequences: The System Dynamic Approach

Predicting the performance of institutions is a challenge. Energy efficiency to meet GHG goals includes a complex set of technological, social, and economic factors. Policy interventions in one part of this system can have unexpected consequences in other parts of the system [70]. Systems dynamics is a tool designed to address such problems, and it has good characteristics for understanding such complex problems [22].

In particular, we represent how public policy in the arenas of energy efficiency and GHG emissions reduction may interplay with corporate sustainability schemes [71]. We foresee that, as a result of the institutional design chosen, unintended consequences may occur should the system become dominated by negative feedback processes resulting in inefficient, worse, or unpredictable outcomes. With the newly approved energy we can expect that implementation will come across difficult challenges and that the expected solution may find "side effects". The side effects, as we foresee, concern the fact that while trying to create a more energy efficient industry with fewer emissions, investment in other sustainable arenas will be reduced, having an overall effect that is worse had firms continued their corporate sustainability strategy [7,63]. This is usually called unintended consequences: the tendency for an intervention to be defeated by the system's response to the intervention itself [30,72].

The system as it is modeled creates feedback loops, a complicated map that helps explain all the decisions that have to be taken in a firm and which not only include the decisions taken by the firm, but the new restrictions imposed by the regulator. Unintended consequences in such a situation may arise when the expectation of a stringent regulation may impair the overall goals laid out by the reform in the first place [73]. The complexity of the number of decisions has to be simulated before decisions in policy are executed.

Firms will be required to make decisions that are complicated to balance, and when considering the analysis derived from the institutional arrangements through Young's analysis, we expect problems to be derived from the change in policies over time. The new regulations have taken time to see the light, and at times the rush of political terms is different to the times when business decisions have to be made. This may affect the overall results. Another aspect is the amount of feedback involved in the decisions made by firms and the regulator. As an example of a single regulatory decision, the Energy Transition Law for firms to have a minimum amount of clean energy in their processes reduces their strategic investment in sustainable practices [12]. To simulate such decision-making processes, we develop a model whose structure is outlined next.

### 4.1. Model Structure

Before introducing a causal loop diagram that will describe the system as a whole, we first present a subsystem diagram in Figure 1 aims to show the model structure in a simple manner. The diagram depicts the primary subsections that influence GHG emissions as a function of production which has strategic investment decisions on clean to non-clean production. The diagram is a high-level guide to a more complicated causal loop diagram [1,24].

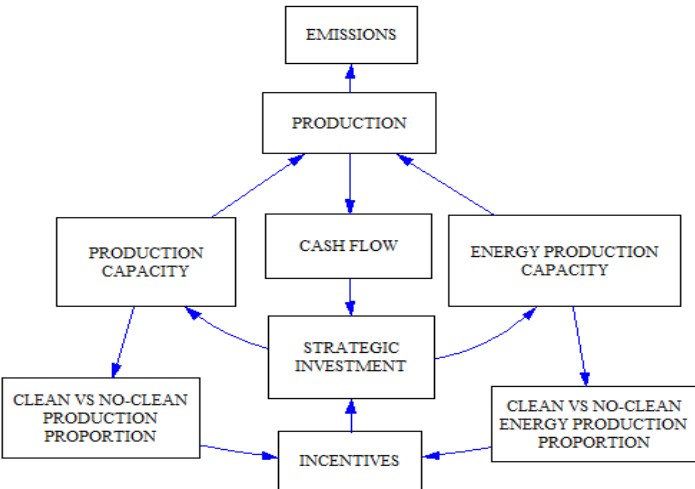

**Figure 1.** Subsystem diagram for the model of clean to non-clean strategic investment.

In our model, strategic investment is a function of two sources: one the internal or organic source that comes from cash flow generated by production and two, the external source which comes from incentives. Incentives are generated by two proportions the clean to non-clean production processes proportion and the clean to non-clean energy production proportion.

Moving into a causal loop diagram, we used Vensim® (Ventana Systems Inc., Harvard, MA, USA) to create the systems model. Figure 2 shows the four loops that constitute the incentives caused by the required proportions of clean production and clean energy generation. Loops 1 and 3 are of the reinforcing loop type (positive loops) because they involve the development of the clean capacity of both the production and generation of energy from direct investments in these areas.

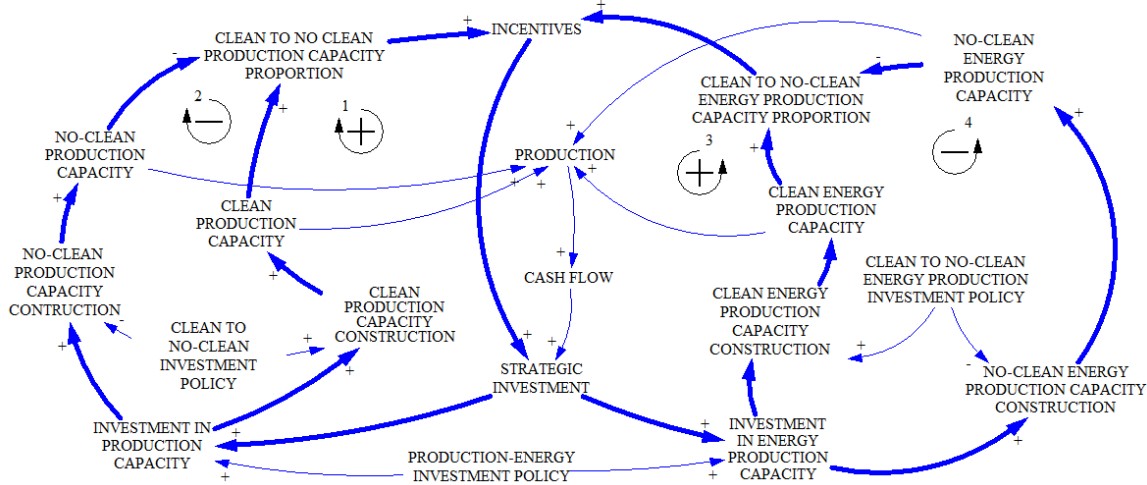

**Figure 2.** Investment incentives causal loop diagrams.

The greater these proportions, the greater the incentives will be. The causal chain for Loop 1 is as follows: the greater the strategic investment, the greater the investment in production capacity, the greater the clean production capacity construction, the greater the clean production capacity, the greater the clean to no-clean production capacity proportion, and the greater the incentives that finally increase the amount of strategic investment, which closes Loop 1. The causal chain for Loop 3 is as follows: the larger the strategic investment, the larger the investment in energy production capacity, the larger the clean energy production capacity construction, the larger the clean energy production capacity, the larger the clean to non-clean energy production capacity proportion and the greater the incentives which ultimately increase incentives, which close Loop 3.

Loops 2 and 4 are of the balancing type (negative loops) since, by increasing the production capacities and the generation of non-clean energies, exogenous support incentives will decrease. The greater these proportions, the greater the incentives will be. The causal chain for Loop 2 is as follows: the larger the strategic investment, the larger the investment in production capacity, the larger the clean production capacity construction, the larger the non-clean production capacity, the smaller the clean to non-clean production capacity proportion, and the lower the incentives that ultimately decrease the amount of strategic investment, which completes Loop 2. The causal chain for Loop 4 is as follows: the higher the strategic investment, the higher the investment in energy production capacity, the higher the non-clean energy production capacity construction, the higher the non-clean energy production capacity, the lower the clean to non-clean energy production capacity proportion, and finally the lower the strategic investment. Thus, to summarize, strategic investment in clean energy under the new law is an incentive-based strategy which means that, unless firms receive subsidies, few will invest out of their pocket directly to meet the new regulations.

The next causal loop diagram, Figure 3, shows the four loops that determine the level of production. The four loops are of the reinforcing type since the greater the investment in each of the two production capacities (clean and non-clean) and the generation of energy (clean and non-clean), the greater the production and, consequently, the cash flow available to increase the subsequent investment will be.

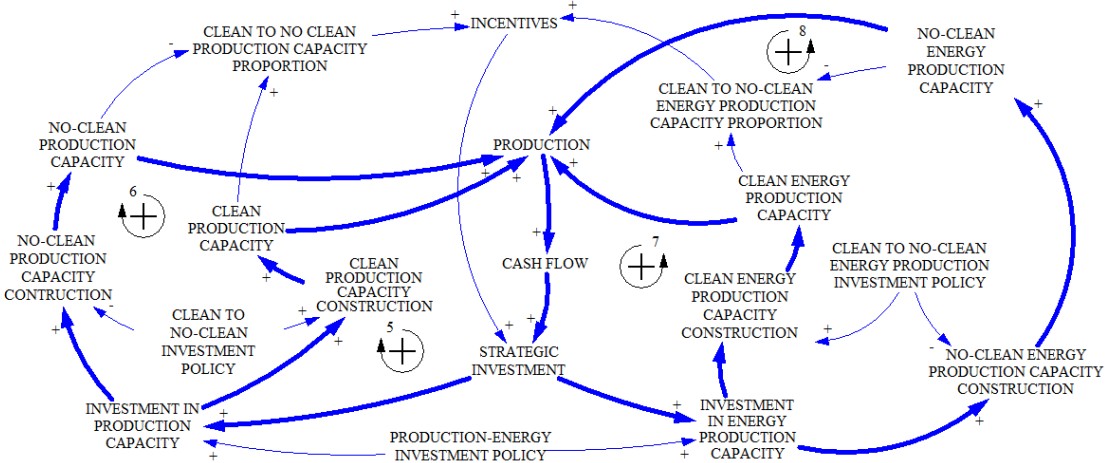

**Figure 3.** Production capacity causal loop diagrams.

When all these loops start to interact together, finding better solutions depends completely on the effects of the system, and these are unclear. In addition, we expect nonlinearity in the decisions taken by firms. In fact, when the reform aims at investment in firms' clean technology or energy investment, the effect on firms' competitiveness can be impacted, reducing the emissions or the need for cleaner technology as stated above, and the overall effect would be that firms will have to find other types of solutions that may impair their current sustainability efforts. As trade-offs by firms appear in the decisions involved, the firm will have to learn to slowly adapt and create new business models that in

the short term may seem to be a solution, but that in the overall long term may test the efficiency of the energy reform. Table 2 shows the variables used, description and the loop component.

**Table 2.** Causal loop variables.

| Variable | Description | Component of Loop: |
|---|---|---|
| PRODUCTION | The level of total production units per year calculated using a Cobb–Douglas type function | 5, 6, 7, 8 |
| CASH FLOW | The level of cash generated per year | 5, 6, 7, 8 |
| STRATEGIC INVESTMENT | The amount of total investment per year, from cash generated and incentives | 1, 2, 3, 4, 5, 6, 7, 8 |
| PRODUCTION VS ENERGY INVESTMENT POLICY | The proportion of production to energy capacity investment policy decision variable | Exogenous variable |
| INVESTMENT IN PRODUCTION CAPACITY | The amount of investment in production capacity per year as a proportion of total investment based of investment production policy | 1, 2, 5, 6 |
| CLEAN VS NO-CLEAN PRODUCTION CAPACITY INVESTMENT POLICY | The proportion of clean to no-clean production capacity investment policy decision variable | Exogenous variable |
| NO-CLEAN PRODUCTION CAPACITY CONSTRUCTION | The amount of no-clean production capacity that is in construction per year | 2, 6 |
| NO-CLEAN PRODUCTION CAPACITY | The amount of operative no-clean production capacity per year | 2, 6 |
| CLEAN PRODUCTION CAPACITY CONSTRUCTION | The amount of clean production capacity that is in construction per year | 1, 5 |
| CLEAN PRODUCTION CAPACITY | The amount of operative clean production capacity per year | 1, 5 |
| CLEAN TO NO CLEAN PRODUCTION CAPACITY PROPORTION | The proportion of clean to no-clean production capacity achieved per year, used as an antecedent to calculate incentives per year | 1, 2 |
| INVESTMENT IN ENERGY PRODUCTION CAPACITY | The amount of investment in energy production capacity per year as a proportion of total investment based of investment energy policy | 3, 4, 7, 8 |
| CLEAN VS NO-CLEAN ENERGY INVESTMENT POLICY | The proportion of clean to no-clean energy production investment policy decision variable | Exogenous variable |
| CLEAN ENERGY PRODUCTION CAPACITY CONSTRUCTION | The amount of clean energy production capacity that is in construction per year | 3, 7 |
| CLEAN ENERGY PRODUCTION CAPACITY | The amount of operative clean energy production capacity per year | 3, 7 |
| NO-CLEAN ENERGY PRODUCTION CAPACITY CONSTRUCTION | The amount of no-clean energy production capacity that is in construction per year | 4, 8 |
| NO-CLEAN ENERGY PRODUCTION CAPACITY | The amount of operative no-clean energy production capacity per year | 4, 8 |
| CLEAN TO NO-CLEAN ENERGY PRODUCTION CAPACITY PROPORTION | The proportion of clean to no-clean energy production capacity used as an antecedent to calculate incentives per year | 3, 4 |
| INCENTIVES | The amount of cash incentives received as a function of the levels of clean to no-clean production and energy production proportions | 1, 2, 3, 4 |

Model validation is important. We use a "causal-descriptive" also called a white box model, relative to a black-box or "correlational" model. In the latter there is no claim of causality in structure and the importance is in the aggregate output behavior of the model often in the form of a classical statistical testing problem and have forecasting properties. The model we use aims to show how real systems actually operates in some aspects, what is relevant is the validity of the internal structure of the model, a theory of a real system that reproduces or predicts behavior and how such behavior is generated [69,74–78].

*4.2. Simulation Analysis*

In this section two scenarios are analyzed: the IMPOSED and the FREE scenarios. The IMPOSED scenario is the one that follows regulations to meet the goals enacted by the government, whereas the FREE scenario allows firms to make their own clean to non-clean investment decisions. The investment strategies for each of the two scenarios are shown in Table 3. The values of the investment proportions in production capacity and power generation, and their corresponding clean to non-clean proportions, are considered in the imposed scenario, based on the Energy Transition Law [54]. In the free scenario, new values of these decision variables are explored, whose impacts improve the key performance indicators (KPIs). That is, the model finds the decision variables based on an optimization of KPIs.

**Table 3.** Scenario assumptions.

|  | Scenarios | |
|---|---|---|
|  | **IMPOSED** | **FREE** |
| % DESIRED ENERGY GENERATION CAPACITY TO PRODUCTION CAPACITY INVESTMENT | 30% | 55% |
| % DESIRED CLEAN TO NON-CLEAN ENERGY PRODUCTION INVESTMENT | 20% | 43% |
| % DESIRED CLEAN TO NOCLEAN PRODUCTION INVESTMENT | 30% | 62% |

The KPIs results under the IMPOSED and FREE scenarios show better results in the latter. Accumulated total emissions are higher in the IMPOSED implying that the government target is met faster under the FREE scenario and the firm's cash flow is higher than that of the IMPOSED scenario. If given the choice, firms will most likely prefer the FREE scenario, a clear corporate sustainability strategy. As far as Incentives KPIs, the regulator will spend slightly less under the IMPOSED scenario but without achieving the overall goal to meet GHG emissions reductions. Table 4 shows the scenario key performance indicators.

**Table 4.** Scenario key performance indicators.

|  | Scenarios | |
|---|---|---|
|  | **IMPOSED** | **FREE** |
| ACCUMULATED TOTAL EMISSIONS INDEX | 4251 | 3761 |
| CASH FLOW PRESENT VALUE | 1152 | 1673 |
| INCENTIVES | 581 | 624 |

The KPIs improve in the free scenario with respect to the imposed scenario. The emissions are lower and as complementary information the cash flow present value is higher and the costs are lower. These values are achieved because the restrictions IMPOSED on official policies, under the FREE scenario are relaxed. Figure 4a,b shows the graphs for the main KPIs of the model: emissions and accumulated emissions that support the argument that the FREE scenario emits fewer emissions than the IMPOSED scenario. These results suggest that the FREE scenario achieves better results not only because of the relaxed restrictions but also because the strategic investment decisions. The strategic investment decision depends on two criteria: The first one is a more convenient allocation policy

of resources and the second is the pace of the strategic investment considering the early benefits of incentives perspective.

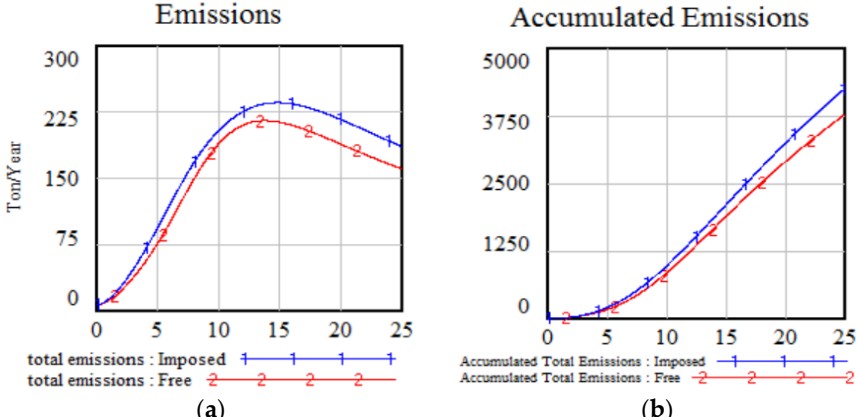

**Figure 4.** (**a**) Emissions; (**b**) Accumulated Emissions.

Figure 5a–d show the cash flow, cash flow present value, total cost, and total cost present value behavior in the scenario analysis. These figures also corroborate the idea that the FREE SCENARIO is more convenient than the IMPOSED scenario.

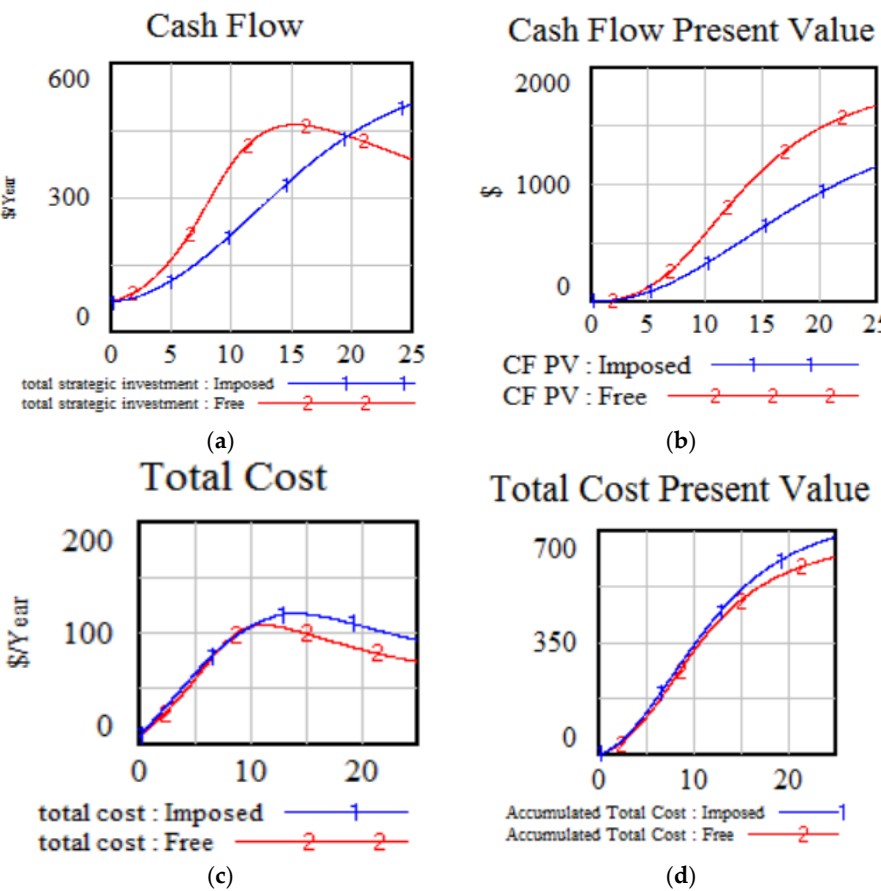

**Figure 5.** Results of the simulation model. (**a**) Cash Flow; (**b**) Cash Flow Present Value; (**c**) Total Cost; (**d**) Total Cost Present Value.

In all cases, the free scenario has a better outcome than the imposed scenario, suggesting that regulation will create unintended consequences. In the next section, we perform a behavior sensitivity analysis to support the formal model validation [74].

### 4.3. Sensitivity Analysis

The sensitivity analysis is a multivariate Monte Carlo analysis [79–81]. We simulate 1000 runs of the model. We assume a uniform distribution for all decision variables. The simulation shows how sensitive emissions are to certain the three decision variables. In Figure 6, total accumulated emissions reference behavior, depend on how much energy to use in the production capacity, how much investment is necessary to spend on clean to non-clean production, and finally the investment of clean to non-clean energy investment.

**Accumulated Total Emissions**

**Figure 6.** Accumulated total emissions reference behavior.

In Figure 7, we can see that there is a wide range of variation of emissions but the overall pattern of behavior is unchanged. Given the constraint of the regulatory restriction by the enacted law, the most cause sensitive decision variable is percentage desired energy generation capacity to production capacity investment which has a relative large range followed in order by, how much investment is necessary to spend on clean to non-clean production and finally the investment of clean to non-clean energy investment.

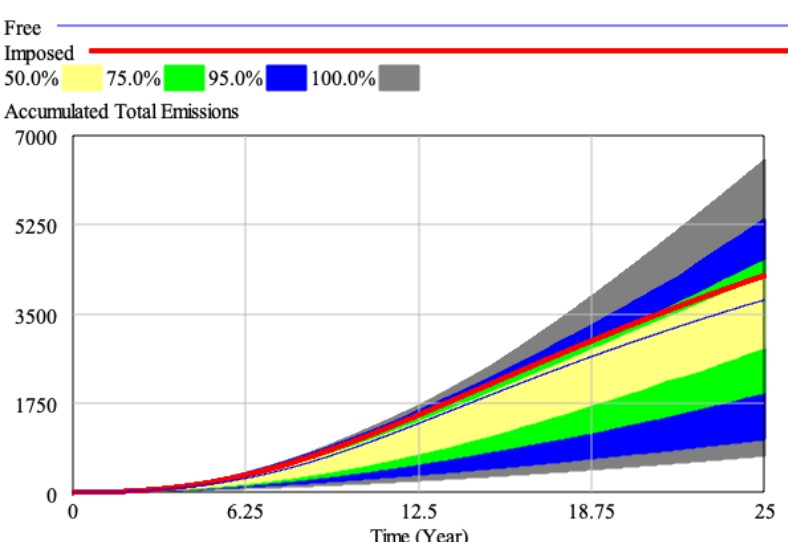

**Figure 7.** Accumulated total emissions multivariate sensitivity analysis.

The sensitivity analysis shows that if the goal is to reduce GHG emissions, the IMPOSED scenario will almost always perform worse the FREE scenario without restrictions. This supports the idea that when you have restrictions, the system structure does not always achieve its best behavior.

## 5. Conclusions

One has to appreciate the optimism that businesses are projecting towards the energy reform. The business sector is coming together with more ambitious and long-term goals, confirming an irreversible trend towards a low-carbon economy. However, goodwill is not enough. For a proper implementation of the reform, the regulators need reliable, verifiable, and updated information, which currently is lacking—and even if it was proposing state-of-the-art regulation, the implementation is behind the reality of the challenge. The main instrument, the National Emission Register (RENE), must be complemented with a clear roadmap on what type of energy technologies will be implemented, by who, and when. In addition, what the carbon price should be and what financial instruments will be considered, and how these will be regulated, should be determined. At the current moment, there is no clarity about these factors.

A low-carbon economy needs mechanisms to facilitate the development of clean technology and the infrastructure to facilitate the development of a market for greenhouse gases and carbon, both of which are not ready yet. Firms and regulators need clear measurement, reports, and verification of emissions. They need a framework for sector-specific actions, as well as a gradual increase in the development of new technologies. Successful mitigation depends in part on access to information and, in some sectors, the development of technology.

Firms have engaged in corporate sustainability strategies and regulation based on targets may impair such sustainability schemes. Firms will have to decide what type of strategic investment to make. As shown by our system dynamic model, a restricted institution will not necessarily meet the Country's target and create unintended consequences.

Energy reform is undoubtedly necessary to transform Mexico, but policy makers have yet to acknowledge one of its great hurdles: implementation itself constitutes a challenge in environmental and economic terms for many actors. Proper implementation of the energy reform in the public sector must harness the power of the business sector and encourage its alignment with the objectives of sustainability. However, if such institutional schemes do not incentivize better sustainable practices than the ones firms currently use, then the energy reform may in fact slow or reverse the growth of sustainable practices in the business sector. If the goal is to reduce GHG emissions, we would suggest to draw on the best practices that have worked in developed countries such as emissions tradable permits that will allow firms to make the most cost effective decisions while meeting GHG reduction targets as well as maintaining their sustainable strategies.

**Author Contributions:** Conceptualization, Methodology, and Analysis A.L.; Formal Analysis, Programming R.D.; Institutional Analysis, M.C.-H.

**Funding:** This research received no external funding.

**Acknowledgments:** The authors wish to acknowledge support by Asociación Mexicana de Cultura, AC for their support in this research and Valeria Dagnino Contreras for her committed assistance to this project.

**Conflicts of Interest:** The authors declare no conflict of interest.

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
