# Peer review of "Regulatory Limits to Corporate Sustainability: How Climate Change Law and Energy Reforms in Mexico May Impair Sustainability Practices in Mexican Firms"

_systems, doi:10.3390/systems7010003_

Round 1
Reviewer 1 Report
First of all, I would like to thank the editors for giving the opportunity to review the article. I appreciate the efforts put in by the authors in developing the manuscript. Development of regulatory regime for climate change is a very new and emerging phenomena. Authors’ attempt in showing the challenges and trade-offs involved in the implementation of climate regulation in Mexico through system modelling approach is praise-worthy.
After reading through manuscript, I do not feel convinced by the arguments put forward by the authors. The paper reads more like a thought experiment than a well thought out rigorously developed journal paper. In the following paragraphs I try to explain my reservations about the quality of the manuscript.
1. The system dynamic model does not seem to have any theoretical or empirical basis. In the descriptive part of the manuscript the authors introduce Young’s policy analysis framework. However, the system dynamic model does not make any use of the Young’s analytical model. The authors describe two models, one with regulation and one without regulation. Young’s model does not imply absence of regulation. Young’s model provides an analytical approach that allows careful considerations of the role of fit, scale and interplay in improving the design and implementation policy prescriptions. A more interesting and robust exercise would have been to compare different models with different configurations of fit, scale and interplays among the several factors of the energy and climate law, many of which the authors have already identified in the descriptive part of the manuscript.
2. No information is provided on the assumptions or the source of data for the results presented in the Table 4 and figure 6.a through to figure 6.f. Thus it is impossible to judge the reliability and validity of these conclusions (apart from the fact that the models have no underlying theoretical or empirical foundations).
3. A key assumption in developing the models is that climate regulations will substitute the existing sustainable activities of the firms and will incur a net cost and will result in net environmental harm. This is a major assumption and not supported by any econometric analysis. Hence, this assumption does not qualify as valid as it is.
4. From the text it appears the authors are trying to highlight only the design flaw and not issues that relates to implementation, since, the entire modelling exercise is conceptual. It does not appear that the authors made any calibration of the models using real data during the implementation phase of the regulations.
5. In section 2 the authors provide some definitions of Causality, performance and design. It is not clear on what basis these definitions are formulated.
6. The authors mention once in page number 2 of the manuscript that they considered best practice recommendation for low-carbon economy from the World Business Council for Sustainable Development (WBCSD). However, it is not clear what these recommendations are, and how these recommendations are implemented in the study. Further, no reference to these so called WBSCD best practice recommendations are included in the manuscript.
7. The referencing is inconsistent.
In summary, unfortunately I am recommending a
reject to publish.
Author Response
México City, December 7, 2018
Dear Reviewer 1:
We appreciate the time and effort you have spent in reviewing our article. We think that your comments are accurate and completely understand your assessment for rejection. My coauthors and I have addressed them all and have put an enormous effort to clarify all your concerns, and amend and correct what was necessary. Broadly speaking we have rewritten large parts of the article as well as included a simulation exercise to justify and strengthen our arguments. In the following paragraphs, I will address each of your questions/comments and describe you exactly how we addressed it and where it the modification in the text. When your suggestions were large, I broke them in shorter comments and answer them accordingly.
Comment 1.
1.1 The system dynamic model does not seem to have any theoretical or empirical basis.
Response 1.1. We have completely revised the section on system dynamics and attended your comments by 1) Addressing the topic of system dynamics in our analysis by supporting our arguments with literature leading scholars in the field (Section 4 page 8), and 2) by adding a discussion and table on the model structure in section 4.1 page 9 that we believe address your theoretical concerns. The empirical basis was addressed by creating only two figures 2 and 3 describing the loops instead of eight figures that were unclear and not well discussed in the original manuscript.
1.2. In the descriptive part of the manuscript the authors introduce Young’s policy analysis framework. However, the system dynamic model does not make any use of the Young’s analytical model.
Response 1.2. We were very careful to rewrite and discuss the institutional analysis, reframe the methodology and support it with additional literature in section 2. We also merged all tables into one that is more specific and deleted those tables that were not adding anything to the manuscript. The system dynamic model now uses one institution, The Energy Transition Law that aims to reduce GHG emissions in 25% by 2018, 30% by 2021. Addressing the scientific question of performance, not design and particularly using the analytical framework of fit. Your suggestions were extremely useful.
1.3 The authors describe two models, one with regulation and one without regulation. Young’s model does not imply absence of regulation.
Response 1.3. You are right, and we made the effort to be clearer to explain that it is not about absence of regulation but rather restriction (i.e. 30% reduction for example). To this end we also address literature about the problem with setting a number as a target and the feasibility to meet it, or commonly known as Goodhart's Law, see : Biber (2008); Newton, (2011) and Reynaert and Sallee, (2016).
1.4 Young’s model provides an analytical approach that allows careful considerations of the role of fit, scale and interplay in improving the design and implementation policy prescriptions. A more interesting and robust exercise would have been to compare different models with different configurations of fit, scale and interplays among the several factors of the energy and climate law, many of which the authors have already identified in the descriptive part of the manuscript.
Response 1.4. We use your comment to include an example in the system dynamic model. The simplest analytical scheme suggested by Young is that of Fit, aiming to help answer the scientific question of institutional performance. The simulation results as well as the sensitivity analysis, a new section on the system dynamic model, used such approach with different values of targets. In all cases, the FREE scenario as we called relative to the IMPOSED scenario, results in better key performance indicators supporting our arguments. The results support the idea that the structure drives the outcomes.
Comment 2. No information is provided on the assumptions or the source of data for the results presented in the Table 4 and figure 6.a through to figure 6.f. Thus it is impossible to judge the reliability and validity of these conclusions (apart from the fact that the models have no underlying theoretical or empirical foundations).
Response 2. The system dynamic approach has been considerably revised to meet your and other tow reviewers´ expectations and comments. It is now Section 4 and it is introduced throughout the paper in the Introduction as well as in the discussion, particularly at the end of Section 3. The method used is properly justified with the literature. We have revised and included a number of papers in the topic of system dynamics, model verification, simulation, and sensitivity analysis by well know authors and leaders in the field. The addition aimed to support our arguments. See Section 4 in page 8 of the manuscript that is now dedicated to the system dynamics approach. We also modified and simplified the tables and figures in the section to convey the message better.
Comment 3. A key assumption in developing the models is that climate regulations will substitute the existing sustainable activities of the firms and will incur a net cost and will result in net environmental harm. This is a major assumption and not supported by any econometric analysis. Hence, this assumption does not qualify as valid as it is.
Response 3. You are right and we did not convey the message properly. We meant to impair or weaken existing activities, not substitute them. There is empirical evidence, based in the literature (Ambec, 2013) that supports the idea that more flexible regulation achieves better results than stringent regulation. There are other incentives as well such as consumers demand or voluntary programs that lead to corporate sustainable strategies, Bansal (2000). Our goal is to suggest that it is the structure of the socio economic systems and the dynamics within the systems and not the outcome what matters most.
Comment 4 From the text it appears the authors are trying to highlight only the design flaw and not issues that relates to implementation, since, the entire modelling exercise is conceptual.
Response 4. Your comment is very valuable and again, you were right. We rewrote the institutional analysis to explain how design is approached within the Energy Reform and how implementation relates to the performance. See Section 2 pages 3 and 4. As far as the modelling exercise that is conceptual, we agree. We want to highland the importance of unintended consequences given the focus on objectives and not in the structure.
Comment 4.2 It does not appear that the authors made any calibration of the models using real data during the implementation phase of the regulations.
Response There is a very large literature on system dynamic calibration where there are pros and cons about using this particular point. We refer to the literature review and our response is based on the literature as follows: Model validation is important. We use a “causal-descriptive” also called a white box model, relative to a black-box or “correlational” model. In the latter there is no claim of causality in structure and the importance lays in the aggregate output behavior of the model often in the form of a classical statistical testing problem and have forecasting properties. The model we use aims to show how real systems actually operates in some aspects, what is relevant is the validity of the internal structure of the model, a theory of a real system that reproduces or predicts behavior and how such behavior is generated. See references by Barlas and Yaman (1996), Barlas et al (1990) Senge and Forrester, (1980) and Sommer (1984)
Comment 5. In section 2 the authors provide some definitions of Causality, performance and design. It is not clear on what basis these definitions are formulated.
We rephrase and rewrote all the scientific questions of performance, causality and design to highlight the analytical schemes of fit, scope and interplay. Then we addressed each part of the new regulation according to these analytical schemes. See section 2, 2.1 to 2.3.
Comment 6. The authors mention once in page number 2 of the manuscript that they considered best practice recommendation for low-carbon economy from the World Business Council for Sustainable Development (WBCSD). However, it is not clear what these recommendations are, and how these recommendations are implemented in the study. Further, no reference to these so called WBSCD best practice recommendations are included in the manuscript.
Response 6. We rewrote the section on corporate sustainability in which we specifically addressed different views on climate change mitigation strategies by firms. We then addressed the WBCSD recommendations and explained them. See Section 3 page 8.
Once again, we appreciate your comments and we hope the new version of the manuscript meets your expectations.
Best Regards,
Antonio Lloret, on behalf of my coauthors Rogerio Domenge and Mildred Castro-Hernández

Reviewer 2 Report
The paper challenges the assumption that regulation aimed at curbing GHG emissions by firms will force firms to face the impact of climate change and create conditions that promote sustainable corporations. Systems dynamic approach by Sterman (2002) was used to model and represent how public policy on energy may interplay with competitive business outcomes. This paper requires minor revision before it is suitable for publication.
Figure 3 (p.8) instruments for the policy on climate change within three orders is distorted. Some text appears to be misisng. This should be rectified.
The transition between sections is not smooth in this paper. For example, section 2.1-2.3 it is unclear what the authors are trying to achieve here. Section 2.2 makes cross reference to Figure 2 but in a very shallow manner there's no elaboration on the role of each actor and it is completely left to the imagination of the reader.
The authors then move into Section 3 on Systems Dynamic Approach abruptly. There needs to be a much stronger justification as to why this approach was chosen over others and also what's out there in the current literature.
In terms of the results, how and why did the authors select scenario assumptions in Table 3? Further and clearer explanation is required.
For section 6, would suggest authors to consider looking at "A Review of Corporate Sustainability Reporting Tools" by Siew which details out the different frameworks, ratings and indices to drive the move towards a cleaner and greener economy and linking this back to their study.
Author Response
México City, December 7, 2018
Dear Reviewer 2:
We appreciate the time and effort you have spent in reviewing our article. We think that your comments are accurate and completely understand your assessment. My coauthors and I have addressed them all and have put an enormous effort to clarify all your concerns, and amend and correct what was necessary. Broadly speaking we have rewritten large parts of the article as well as included a simulation exercise and sensitivity analysis to justify and strengthen our arguments. In the following paragraphs, I will address each of your questions/comments and describe to you exactly how we addressed it and where the modification in the text are. When your suggestions were large, I broke them in shorter comments and answer them accordingly.
Comment 1. Figure 3 (p.8) instruments for the policy on climate change within three orders is distorted. Some text appears to be missing. This should be rectified.
Response 1. We have rectified and summarized all tables in the document regarding the institutional framework into one table. It should be clearer now. Table 1 in Section 2.3 page 5, discusses the implications of the table within the discussion of the institutional analysis.
Comment 2. The transition between sections is not smooth in this paper. For example, section 2.1-2.3 it is unclear what the authors are trying to achieve here.
Response 2. We have rewritten the Institutional analysis framework to address your comments as well as other reviewers. The new analysis makes smoother transitions and it is linked to other sections in the paper.
Comment 3. Section 2.2 makes cross reference to Figure 2 but in a very shallow manner there's no elaboration on the role of each actor and it is completely left to the imagination of the reader.
Response 3. Similarly to Response 1, we have merged all tables related to the institutional framework into one single table that explains actors. Table 1 in Section 2.3 page 5 discusses the implications of the table within the discussion of the institutional analysis.
Comment 4. The authors then move into Section 3 on Systems Dynamic Approach abruptly. There needs to be a much stronger justification as to why this approach was chosen over others and also what's out there in the current literature.
Response 4. The system dynamic approach has been considerably revised to meet the reviewer’s expectations and comments. It is now Section 4 and it is introduced throughout the paper in the Introduction as well as in the discussion, particularly at the end of Section 3. The method used is properly justified with the literature. We have revised and included a number of papers in the topic of system dynamics, model verification, simulation, sensitivity analysis by well-know authors and leaders of the field, to support our arguments. Section 4 in page 8 of the manuscript is dedicated to the system dynamics approach.
Comment 5. In terms of the results, how and why did the authors select scenario assumptions in Table 3? Further and clearer explanation is required.
Response 5. Your recommendations on the scenario assumptions in section 4.2 as well as the simulation analysis can be found in section 4.3. All the assumptions are supported by the system dynamics literature as well.
Comment 6. For section 6, would suggest authors to consider looking at "A Review of Corporate Sustainability Reporting Tools" by Siew which details out the different frameworks, ratings and indices to drive the move towards a cleaner and greener economy and linking this back to their study.
Response 6. We have considered your recommendations on Siew (2015) regarding corporate sustainability reporting tools. We have also rewritten large parts of the corporate sustainability section to make the document to flow easier. We have also moved the corporate sustainability section before the systems dynamic approach, a suggestion by Reviewer 3. In addition we extended the analysis of the topic to include leading authors. The section, New Challenges for Firms: Corporate Sustainability within the Energy Reform is now under Section 3.
Once again, we appreciate your comments and we hope the new version of the manuscript meets your expectations.
Best Regards,
Antonio Lloret, on behalf of my coauthors Rogerio Domenge and Mildred Castro-Hernández

Reviewer 3 Report
The paper challenges the assumption that state of the art regulation aimed at curbing GHG emissions is the panacea that will force firms to face the impact of climate change and create conditions that promote sustainable corporations. The method deployed is scenario analysis. The authors find that the institutional design analysed is inefficient in triggering desired change.
The topic is highly relevant and worthy of investigation. Unfortunately however the paper is poorly written so reading this paper was not a pleasure at all. To be honest, my impression is that this manuscript is a failed attempt to convert a project report into an academic paper. Furthermore, I suspect that this paper has been around for a long time and has witnessed many rejections (see comment 2).
Some suggestions:
1) The aim of the paper. Not clear at all!! I must have reviewed over 100 papers in my career yet I do not recall a paper where the aim is not stated clearly in the introduction section. The closest to the aim are the first two sentences in the abstract, but why can the authors not state clearly what is the aim of the paper (both in the abstract and the introduction section). I infer that the authors wish to demonstrate that Mexican regulation is ineffective in bringing about desired corporate change but I could also be wrong on this. It is author’s duty to convey this message clearly to the reader.
2) Outdated data. Authors report forecasted GDP data for 2013 and 2014 at the start of the paper!! We are now in 2018! Have the authors changed nothing to the paper in the last 5 years??
3) Theory and propositions. What is/are the proposition(s) of this paper?? Three statements in italic text on page 3 appear to be an attempt to formulate propositions, yet these are not propositions. See any textbook on theory building how to develop propositions. An attempt to formulate a proposition can be found on page 10, paragraph 1 (we foresee that….). The authors however provide absolutely no rationale for their “propositions”. Why are the authors’ not absolutely clear what is it about the chosen institutional design that is likely to trigger unintended consequences?
4) Tables. Overly long and not informative. Why do you need such long tables? Can you really not do a better job conveying more condense information to the reader?
5) Model structure. I do not understand the model nor how it has been derived. In fact, due to my poor vision I cannot even read the figures as the writing font is too small. I suspect that the model includes many variables and relations among them. How were these variables selected and how were relations between them developed?
6) Scenarios. I do not understand how the scenarios were developed nor their relation to the model.
7) Section 6 appears to be a literature review. It is very awkward to position a literature review at the end of the paper. Literature review should be positioned at the beginning of the paper. Literature review is also outdated. Add some recent relevant papers to the literature, e.g. Cadez S., Czerny, A., Letmathe, P. (in press): Stakeholder pressures and corporate climate change mitigation strategies. Business Strategy and the Environment, Cadez S., Guilding C. (2017): Examining distinct carbon cost structures and climate change abatement strategies in CO2 polluting firms. Accounting, Auditing & Accountability Journal, 30, 1041-1064, Cadez S., Czerny A. (2016): Climate change strategies in carbon intensive firms. Journal of cleaner production, 112, 4132-4143.
8) Conclusion. Only 3 sentences and statements so broad that one could have written these without any study being conducted to support them. Be absolutely clear which findings are a direct result of this study and position your findings relative to prior papers in the area (e.g. those listed in comment 7).
Author Response
México City, December 7, 2018
Dear Reviewer 3:
We appreciate the time and effort you have spent in reviewing our article. We think that your comments are accurate and completely understand your assessment. My coauthors and I have addressed them all and have put an enormous effort to clarify all your concerns, and amend and correct what was necessary. Broadly speaking we have rewritten large parts of the article as well as included a simulation exercise and sensitivity analysis to justify and strengthen our arguments. In the following paragraphs, I will address each of your questions/comments and describe to you exactly how we addressed it and where the modifications in the text are. When your suggestions were large, I broke them in shorter comments and answer them accordingly.
Comments:
The topic is highly relevant and worthy of investigation. Unfortunately however the paper is poorly written so reading this paper was not a pleasure at all. To be honest, my impression is that this manuscript is a failed attempt to convert a project report into an academic paper. Furthermore, I suspect that this paper has been around for a long time and has witnessed many rejections (see comment 2).
Response. The paper has never been submitted to a journal before. The story of the paper started back when the Energy Transition Law was passed. In a breakfast with business and industrial executives, it was suggested that should the energy transition law passed, the firm will have to substitute current practices to meet the new regulation. At that point the paper was generated and a first draft of the paper was presented in Paris at CERALE conference in June 2016. Then we started working on the system dynamic model and the revised version was presented in a workshop at the Academy of Management in Chicago 2018. The latter feedback came with mix comments but was considered an insightful question. Thus we pursue publication.
Comment 1. The aim of the paper. Not clear at all!! I must have reviewed over 100 papers in my career yet I do not recall a paper where the aim is not stated clearly in the introduction section. The closest to the aim are the first two sentences in the abstract, but why can the authors not state clearly what is the aim of the paper (both in the abstract and the introduction section). I infer that the authors wish to demonstrate that Mexican regulation is ineffective in bringing about desired corporate change but I could also be wrong on this. It is author’s duty to convey this message clearly to the reader.
Response 1: We hope that while revising and rewriting each section of the paper, our message is clearer. Our main message is to show that sustainable behavior by firms may be impaired by regulatory restrictions. We challenge the assumption that regulation aimed at curbing greenhouse gas emissions (GHG) on the form of a target to meet the Country’s GHG emissions commitments will promote sustainable corporations. We argue that, in fact, such regulation may impair sustainability practices because it creates unintended consequences. The idea is developed throughout the document and supported in the system dynamic model discussion.
2. Outdated data. Authors report forecasted GDP data for 2013 and 2014 at the start of the paper!! We are now in 2018! Have the authors changed nothing to the paper in the last 5 years??
Response 2: You are right, thank you. We revised all data that was necessary and deleted the data that was not relevant for the development of the argument. Some parts of the energy reform started in 2013 and most of the regulation was developed in the next two years. We have used current data as of April 2018 to show the early effects of the Energy Reform.
3. Theory and propositions. What is/are the proposition(s) of this paper?? Three statements in italic text on page 3 appear to be an attempt to formulate propositions, yet these are not propositions. See any textbook on theory building how to develop propositions. An attempt to formulate a proposition can be found on page 10, paragraph 1 (we foresee that....). The authors however provide absolutely no rationale for their “propositions”. Why are the authors’ not absolutely clear what is it about the chosen institutional design that is likely to trigger unintended consequences?
Response 3. Your comment is very valuable and again, you were right. We rewrote the institutional analysis to explain how design is approached within the Energy Reform and how implementation relates to the performance. See Section 2 pages 3 and 4. We are not using propositions anymore but rather having a discussion of the analytical frameworks. We spent more time in supporting the system dynamic model that now models one institutional scheme.
4. Tables. Overly long and not informative. Why do you need such long tables? Can you really not do a better job conveying more condense information to the reader?
Response 4: We have revised the tables in order to compress them into only one more informative table regarding the institutional framework with all the necessary data and eliminate the ones we did not need, utilizing the information on the paper. Table 1 in Section 2.3 page 5, discusses the implications of the table within the discussion of the institutional analysis.
5. Model structure. I do not understand the model nor how it has been derived. In fact, due to my poor vision I cannot even read the figures as the writing font is too small. I suspect that the model includes many variables and relations among them. How were these variables selected and how were relations between them developed?
Response 5: The system dynamic approach has been considerably revised to meet the reviewer’s expectations and comments. It is now Section 4 and it is introduced throughout the paper in the Introduction as well as in the discussion, particularly at the end of Section 3. The method used is properly justified with the literature. We have revised and included a number of papers in the topic of system dynamics, model verification, simulation, sensitivity analysis by well-know authors and leaders of the field, to support our arguments. Section 4 in page 8 of the manuscript is dedicated to the system dynamics approach. We have made changes to the model images for an easier comprehension, eliminated the figures that were not necessary.
6. Scenarios. I do not understand how the scenarios were developed nor their relation to the model.
Response 6. We tackled your recommendations on the scenario assumptions in section 4.2 as well as the simulation analysis that can be found in section 4.3. All the assumptions are supported by the system dynamics literature as well.
7. Section 6 appears to be a literature review. It is very awkward to position a literature review at the end of the paper. Literature review should be positioned at the beginning of the paper. Literature review is also outdated. Add some recent relevant papers to the literature, e.g. Cadez S., Czerny, A., Letmathe, P. (in press): Stakeholder pressures and corporate climate change mitigation strategies. Business Strategy and the Environment, Cadez S., Guilding C. (2017): Examining distinct carbon cost structures and climate change abatement strategies in CO2 polluting firms. Accounting, Auditing & Accountability Journal, 30, 1041-1064, Cadez S., Czerny A. (2016): Climate change strategies in carbon intensive firms. Journal of cleaner production, 112, 4132-4143.
Response 7: We have moved Section 6, New Challenges for Firms: Corporate Sustainability within the Energy Reform, to section 3. Upon your suggestion, we revised its place in the document overall and moved it to a place where it support our arguments.. Furthermore, we have studied the suggested literature and incorporated it into our review, not only the suggested ones such as Cadez and Czerny (2016), Cadez and Guilding (2017), and Cadez (2018); but also Lyon (2018), Siew (2015), and González (2015).
8. Conclusion. Only 3 sentences and statements so broad that one could have written these without any study being conducted to support them. Be absolutely clear which findings are a direct result of this study and position your findings relative to prior papers in the area (e.g. those listed in comment 7).
Response 8: You were right, thank you. Our conclusion was not specific. We have written a better conclusion which explains everything that we have done on this paper, from our research question, our methodology, our strategy and especially the results from our analysis. We have explained the importance of our findings and highlighted the importance of unintended consequences when the focused is on objectives rather than on the structure. The conclusion in section 8 is now a few paragraphs long with resonance from the whole paper.
Once again, we appreciate your comments and we hope the new version of the manuscript meets your expectations.
Best Regards,
Antonio Lloret, on behalf of my coauthors Rogerio Domenge and Mildred Castro-Hernández

Round 2
Reviewer 1 Report
I appreciate authors' attempt at revising the manuscript, however the revisions does not adequately address the concerns I raised in my first review. Hence, unfortunately I have to again reject the paper.
Author Response
México City, January 6, 2018
Dear Reviewer 1:
Thank you for reading the article again, however we are not sure what specific responses were inadequate. Perhaps your guidance will help us answer them better. In our reading of your comments on the first review, there were three main domains that needed clarification, these are:
1) The institutional analysis
2) The system dynamic approach and,
3) The simulation analysis.
Using the first review responses as a guide, the way that we approach each domain is as follows:
1) The institutional analysis. We have focused on supporting our arguments to show that implementation of an institution is complicated because there are analytical issues to attend. These are Fit, Scale and Interplay. We used the literature to rephrase all the institutional analysis. We focused our analysis on the “Fit” element, per your suggestion to focus on only one issue (comment 1.4 of First Review). The Mexican Energy Reform and Climate Change Law aims to reduce GHG emissions and aims to do so particularly in firms by an Energy Transition Law that aims to use Clean Energy as the main strategy for firms GHG emissions reductions. Responses 1.2, 1.3, 1.4 tackle your concerns in detail.
2) The system dynamic approach. We understand your concern that the system dynamic is a conceptual model or theoretical model and that it lacks empirical evidence. The reason to use a system dynamic is precisely to show that policy interventions in one part of this system can have unexpected consequences in other parts of the system. We used main authors and literature on systems topic to support the idea that systems dynamics is a tool designed to address such problems, and it has good characteristics for understanding such complex problems. We have included a Figure 1 to show the conceptual structure of the system dynamics. Responses 1.1, 1.2, 3 and 4, tackle your concerns in detail.
3) The simulation analysis. We have clarified assumptions and the description of the model to show how the overall structure of the model works. We understand your concern about assumptions and calibration. To this end we have done an extensive literature review on how to better present the methodology and model conceptualization and we have included a new section on sensitivity analysis that show that the way the Energy Transition Law is will have unintended consequences in firms sustainable strategy. Responses 2, 3 and 4, tackle your concerns in detail.
We believe that our responses on the first review have addressed all your concerns and we appreciate that your review has strengthen our manuscript.
Best Regards,
Antonio Lloret on behalf of coauthors.
Below is our original response to your first review.
We appreciate the time and effort you have spent in reviewing our article. We think that your comments are accurate and completely understand your assessment for rejection. My coauthors and I have addressed them all and have put an enormous effort to clarify all your concerns, and amend and correct what was necessary. Broadly speaking we have rewritten large parts of the article as well as included a simulation exercise to justify and strengthen our arguments. In the following paragraphs, I will address each of your questions/comments and describe you exactly how we addressed it and where it the modification in the text. When your suggestions were large, I broke them in shorter comments and answer them accordingly.
Comment 1.
1.1 The system dynamic model does not seem to have any theoretical or empirical basis.
Response 1.1. We have completely revised the section on system dynamics and attended your comments by 1) Addressing the topic of system dynamics in our analysis by supporting our arguments with literature leading scholars in the field (Section 4 page 8), and 2) by adding a discussion and table on the model structure in section 4.1 page 9 that we believe address your theoretical concerns. The empirical basis was addressed by creating only two figures 2 and 3 describing the loops instead of eight figures that were unclear and not well discussed in the original manuscript.
1.2. In the descriptive part of the manuscript the authors introduce Young’s policy analysis framework. However, the system dynamic model does not make any use of the Young’s analytical model.
Response 1.2. We were very careful to rewrite and discuss the institutional analysis, reframe the methodology and support it with additional literature in section 2. We also merged all tables into one that is more specific and deleted those tables that were not adding anything to the manuscript. The system dynamic model now uses one institution, The Energy Transition Law that aims to reduce GHG emissions in 25% by 2018, 30% by 2021. Addressing the scientific question of performance, not design and particularly using the analytical framework of fit. Your suggestions were extremely useful.
1.3 The authors describe two models, one with regulation and one without regulation. Young’s model does not imply absence of regulation.
Response 1.3. You are right, and we made the effort to be clearer to explain that it is not about absence of regulation but rather restriction (i.e. 30% reduction for example). To this end we also address literature about the problem with setting a number as a target and the feasibility to meet it, or commonly known as Goodhart's Law, see : Biber (2008); Newton, (2011) and Reynaert and Sallee, (2016).
1.4 Young’s model provides an analytical approach that allows careful considerations of the role of fit, scale and interplay in improving the design and implementation policy prescriptions. A more interesting and robust exercise would have been to compare different models with different configurations of fit, scale and interplays among the several factors of the energy and climate law, many of which the authors have already identified in the descriptive part of the manuscript.
Response 1.4. We use your comment to include an example in the system dynamic model. The simplest analytical scheme suggested by Young is that of Fit, aiming to help answer the scientific question of institutional performance. The simulation results as well as the sensitivity analysis, a new section on the system dynamic model, used such approach with different values of targets. In all cases, the FREE scenario as we called relative to the IMPOSED scenario, results in better key performance indicators supporting our arguments. The results support the idea that the structure drives the outcomes.
Comment 2. No information is provided on the assumptions or the source of data for the results presented in the Table 4 and figure 6.a through to figure 6.f. Thus it is impossible to judge the reliability and validity of these conclusions (apart from the fact that the models have no underlying theoretical or empirical foundations).
Response 2. The system dynamic approach has been considerably revised to meet your and other tow reviewers´ expectations and comments. It is now Section 4 and it is introduced throughout the paper in the Introduction as well as in the discussion, particularly at the end of Section 3. The method used is properly justified with the literature. We have revised and included a number of papers in the topic of system dynamics, model verification, simulation, and sensitivity analysis by well know authors and leaders in the field. The addition aimed to support our arguments. See Section 4 in page 8 of the manuscript that is now dedicated to the system dynamics approach. We also modified and simplified the tables and figures in the section to convey the message better.
Comment 3. A key assumption in developing the models is that climate regulations will substitute the existing sustainable activities of the firms and will incur a net cost and will result in net environmental harm. This is a major assumption and not supported by any econometric analysis. Hence, this assumption does not qualify as valid as it is.
Response 3. You are right and we did not convey the message properly. We meant to impair or weaken existing activities, not substitute them. There is empirical evidence, based in the literature (Ambec, 2013) that supports the idea that more flexible regulation achieves better results than stringent regulation. There are other incentives as well such as consumers demand or voluntary programs that lead to corporate sustainable strategies, Bansal (2000). Our goal is to suggest that it is the structure of the socio economic systems and the dynamics within the systems and not the outcome what matters most.
Comment 4 From the text it appears the authors are trying to highlight only the design flaw and not issues that relates to implementation, since, the entire modelling exercise is conceptual.
Response 4. Your comment is very valuable and again, you were right. We rewrote the institutional analysis to explain how design is approached within the Energy Reform and how implementation relates to the performance. See Section 2 pages 3 and 4. As far as the modelling exercise that is conceptual, we agree. We want to highland the importance of unintended consequences given the focus on objectives and not in the structure.
Comment 4.2 It does not appear that the authors made any calibration of the models using real data during the implementation phase of the regulations.
Response There is a very large literature on system dynamic calibration where there are pros and cons about using this particular point. We refer to the literature review and our response is based on the literature as follows: Model validation is important. We use a “causal-descriptive” also called a white box model, relative to a black-box or “correlational” model. In the latter there is no claim of causality in structure and the importance lays in the aggregate output behavior of the model often in the form of a classical statistical testing problem and have forecasting properties. The model we use aims to show how real systems actually operates in some aspects, what is relevant is the validity of the internal structure of the model, a theory of a real system that reproduces or predicts behavior and how such behavior is generated. See references by Barlas and Yaman (1996), Barlas et al (1990) Senge and Forrester, (1980) and Sommer (1984)
Comment 5. In section 2 the authors provide some definitions of Causality, performance and design. It is not clear on what basis these definitions are formulated.
Response 5. We rephrase and rewrote all the scientific questions of performance, causality and design to highlight the analytical schemes of fit, scope and interplay. Then we addressed each part of the new regulation according to these analytical schemes. See section 2, 2.1 to 2.3.
Comment 6. The authors mention once in page number 2 of the manuscript that they considered best practice recommendation for low-carbon economy from the World Business Council for Sustainable Development (WBCSD). However, it is not clear what these recommendations are, and how these recommendations are implemented in the study. Further, no reference to these so called WBSCD best practice recommendations are included in the manuscript.
Response 6. We rewrote the section on corporate sustainability in which we specifically addressed different views on climate change mitigation strategies by firms. We then addressed the WBCSD recommendations and explained them. See Section 3 page 8.
Once again, we appreciate your comments and we hope the new version of the manuscript meets your expectations.
Best Regards,
Antonio Lloret, on behalf of my coauthors Rogerio Domenge and Mildred Castro-Hernández
Reviewer 2 Report
Thanks for addressing all the comments. The paper is now ready for publication in its current form.
Author Response
Dear Reviewer 2,
Thank you for your feedback. We are grateful for your comments and we have now checked the manuscript for English language and spelling.
Best Regards,
Antonio Lloret on behalf of coauthors.
Reviewer 3 Report
The authors have addressed the concerns expressed in the first review fairly well. The paper now reads more logically, the arguments and conclusions are presented more clearly (albeit not yet completely clear). Please note that many readers potentially interested in this paper will not be familiar with the method deployed of this paper, but every reader should be able to understand the introduction, the rationale and the conclusions of the paper irrespective of his/her command of the method.
In order to increase clarity, I suggest a few minor changes to better convey the message of this paper to the readers.
1. Abstract. Should convey the key message(s) on a stand-alone basis. In the last sentence the authors explicate that »as a result of the institutional scheme chosen…«. At this point the reader becomes interested what are the characteristics of the scheme therefore I would add here (maybe in parentheses) these key characteristics. I take it from the paper that the key characteristics are (1) the complexity and interdependence of regulations on different levels (from national to municipal), (2) initial problems with setting up the national GHG emissions registry for carbon intensive firms, (3) lack of national market for GHG emission allowances. If I got these key characteristics wrong it is probably authors fault for not explicating them more clearly.
2. Rather than just providing criticism, I would also like to see some optimistic tone in the paper. It seems to me that Mexico is now at a stage where European Union was some 20 years ago. At the start of the climate change agenda, EU has had very similar problems to those identified in comment 1 but has gradually solved them rather successfully. EU has enacted common EU legislation, implemented an EU wide registry of GHG emissions and launched an EU Emissions Trading System (EU ETS) in 2005. At the start of the scheme, companies were unsure how to manage their GHG emissions to cope with the scheme (please cite a paper by Cadez and Czerny, 2010, Carbon management strategies in manufacturing companies: An exploratory note; Journal for East European Management Studies) but as time progressed it became evident that EU ETS emissions (more precisely the emissions of companies in the EU ETS) are successfully decreasing in time (see https://www.eea.europa.eu/data-and-maps/dashboards/emissions-trading-viewer-1). To summarize, I do not share the same level of pessimism with the authors, in particular because the authors compare only two scenarios from an indefinite number of likely scenarios. The problems identified by the authors are typical for any new system but are likely to be solved later when time progresses.
3. Related to comment 2, it would also be good to conclude the paper with some constructive implications. Building on comment 2, maybe the authors could propose that Mexico should simply adopt the characteristics of the systems that work somewhere else, such as the EU ETS?
Author Response
Dear Reviewer 3,
Thank you for your constructive feedback. Below is a detailed description of how we have incorporated your comments into the manuscript. Additionally, we have revised the manuscript for English language and style.
General comment 0.1. The authors have addressed the concerns expressed in the first review fairly well. The paper now reads more logically, the arguments and conclusions are presented more clearly (albeit not yet completely clear).
· Response 0.1. Thank you. Your comments and those of the other reviewers during the first round were extremely useful. We have also revised the paper for English and language style to make it clearer.
General Comment 0.2. Please note that many readers potentially interested in this paper will not be familiar with the method deployed of this paper, but every reader should be able to understand the introduction, the rationale and the conclusions of the paper irrespective of his/her command of the method.
· Response 0.2. We have tried to make clear the methodology and its application by supporting it in the literature. We have also shared the manuscript with a couple of colleagues from other departments at our university in order to get feedback from readers of different academic background.
Now, moving into your detailed comments:
Comment 1.1. Abstract. Should convey the key message(s) on a stand-alone basis. In the last sentence the authors explicate that »as a result of the institutional scheme chosen…«. At this point the reader becomes interested what are the characteristics of the scheme therefore I would add here (maybe in parentheses) these key characteristics.
· Response 1.1. We clarify in the abstract what the characteristics of the institutional scheme are as an addition to the sentence. In particular in the introduction we clarify that such scheme is: One of the main institutions enacted was a new clean energy policy, its Energy Transition Law enacted in December 2015, the law includes a clean energy target: 25% of electricity generation by 2018, 30% by 2021, and 35% by 2024, these targets have been relevant to shape GHG emissions trajectories.
· Later in the last paragraph of the introduction, we change the wording “Mexican regulation” to Energy Transition Law that is specifically the institutional scheme that we model and simulate later in the manuscript. Now it reads, ”…In order to address these complex relationships, we consider two important elements that are worth addressing for an effective implementation. First, we aim at the national-wide institutional infrastructure and its effect on firms; and second, the non-market alternative of corporate sustainability and how it must be considered to reach the NDC goals. We tackle these two elements to show that the Energy Transition Law aimed at curving GHG emissions on the form of a clean energy law may prove to be ineffective and that its effects may impair the current corporate sustainability practices of Mexican firms.”
Comment 1.2 I take it from the paper that the key characteristics are (1) the complexity and interdependence of regulations on different levels (from national to municipal), (2) initial problems with setting up the national GHG emissions registry for carbon intensive firms, (3) lack of national market for GHG emission allowances. If I got these key characteristics wrong it is probably authors fault for not explicating them more clearly.
· Response 1.2. The key characteristics that you have mentioned are correct. I will add one more: (4) Setting a target or an objective measure without considering the structure may result in unintended consequences.
Comment 2.1. Rather than just providing criticism, I would also like to see some optimistic tone in the paper. It seems to me that Mexico is now at a stage where European Union was some 20 years ago. At the start of the climate change agenda, EU has had very similar problems to those identified in comment 1 but has gradually solved them rather successfully. EU has enacted common EU legislation, implemented an EU wide registry of GHG emissions and launched an EU Emissions Trading System (EU ETS) in 2005. At the start of the scheme, companies were unsure how to manage their GHG emissions to cope with the scheme (please cite a paper by Cadez and Czerny, 2010, Carbon management strategies in manufacturing companies: An exploratory note; Journal for East European Management Studies) but as time progressed it became evident that EU ETS emissions (more precisely the emissions of companies in the EU ETS) are successfully decreasing in time (seehttps://www.eea.europa.eu/data-and-maps/dashboards/emissions-trading-viewer-1).
· Response 2.1. Thank you for guiding us to Cadez and Czerny, 2010. It is important to mention though that the current Energy Transition Law aims for firms to use Clean Energy and the amount has been set to target use energy, not emissions, with targets of a 25% by 2018, 30% by 2021, and 35% by 2024. These targets are relevant to shape GHG, but it is not straightforward that the level of emissions will go down in the same amount as the clean energy usage. Take a production intensive firm that regardless of the energy source has GHG emissions of X as a by-product. Changes in the energy strategy certainly will impact total emissions by the firm (production plus energy sources) but an alternative strategy may be to change only production. The level of emissions may be lower under the alternative strategy than on the regulatory strategy. This specific situation is simulated in the strategic dynamic exercise. An alternative will be the one that Cadez and Czerny, 2010 mentions in terms of the inherent flexibility of the emissions allowances. As the authors mention: “It offers companies the possibility of tailoring a carbon management strategy that is the most cost-effective, i.e. reducing actual emissions vs. buying allowances to emit.” We have included a paragraph on the paper suggested in page 2 third paragraph.
Comment 2.2. To summarize, I do not share the same level of pessimism with the authors, in particular because the authors compare only two scenarios from an indefinite number of likely scenarios. The problems identified by the authors are typical for any new system but are likely to be solved later when time progresses.
· Response 2.2. We understand your point and appreciate it; however, we want to underline the fact that setting a target may have unintended consequences.
Comment 3. Related to comment 2, it would also be good to conclude the paper with some constructive implications. Building on comment 2, maybe the authors could propose that Mexico should simply adopt the characteristics of the systems that work somewhere else, such as the EU ETS?
· Response 3. Thank you, we have included a sentence in the last paragraph following your suggestions.
Best Regards,
Antonio Lloret on behalf of coauthors.
